# Spatiotemporal Forecasting as Planning: A Model-Based Reinforcement Learning Approach with Generative World Models

## Abstract

Physical spatiotemporal forecasting poses a dual challenge: The inherent stochasticity of physical systems makes it difficult to capture extreme or rare events, especially under *data scarcity*. Moreover, many critical domain-specific metrics are *non-differentiable*, precluding their direct optimization by conventional deep learning models. To address these challenges, we introduce a new paradigm, *Spatiotemporal Forecasting as Planning*, and propose SFP, a framework grounded in Model-Based Reinforcement Learning. First, SFP constructs a novel Generative World Model to learn and simulate the physical dynamics system. This world model comprises a deterministic base network and a probabilistic Multi-scale Top-K Vector Quantized decoder. It not only provides a single-point prediction of the future but also generates a distribution of diverse, high-fidelity future states, enabling "imagination-based" simulation of the environment's evolution. Building on this foundation, the base forecasting model acts as an *Agent*, whose output is treated as an action to guide exploration. We then introduce a *Planning Algorithm based on Beam Search*. This algorithm performs forward exploration within the learned world model, leveraging the non-differentiable domain metrics as a *Reward Signal* to identify high-return future sequences. Finally, these high-reward candidates, identified through planning, serve as high-quality pseudo-labels to continuously optimize the agent's *Policy* through an iterative self-training process. The SFP framework seamlessly integrates world model learning with reward-based planning, fundamentally addressing the challenge of optimizing non-differentiable objectives and mitigating data scarcity via exploration in its internal simulations. Comprehensive experiments on multiple benchmarks show that SFP not only significantly reduces prediction error (e.g., up to 39% MSE reduction) but also demonstrates exceptional performance on critical domain metrics, including physical consistency and the ability to capture extreme events. Our codes are available at https://github.com/easylearningscores/SFP.

## 1 Introduction

Spatio-temporal forecasting serves as a cornerstone of modern science and engineering, playing an indispensable role in critical domains ranging from high-impact weather alerts and long-term climate modeling to fluid dynamics analysis in aerospace engineering (Wu et al., 2025; Gao et al., 2025; Bi et al., 2023; Wu et al., 2024b; Lam et al., 2023). In recent years, with the remarkable rise of deep learning, data-driven approaches, particularly models based on Convolutional Neural Networks (CNNs) (Shi et al., 2015; Raonic et al., 2023; Gao et al., 2022a), Transformers (Gao et al., 2022b; Wu et al., 2024a), and Neural Operators (Li et al., 2020; Wu et al., 2024d; Bonev et al., 2023), have demonstrated exceptional capabilities. They efficiently learn complex, nonlinear dynamics from high-dimensional spatio-temporal data, often surpassing traditional, computationally expensive numerical simulations in both prediction efficiency and accuracy on many benchmarks. This series of breakthroughs is ushering AI for Science into a new era of immense possibilities, promising an unprecedented enhancement in our ability to understand and predict the complex physical world.

However, despite these remarkable successes, the vast majority of current data-driven forecasting models operate on a fundamentally flawed assumption: that optimizing simple, pixel-wise proxy

losses, such as Mean Squared Error (MSE) (Gao et al., 2022a; Wu et al., 2024c; Schneider et al., 2017), is sufficient to achieve superior real-world performance. This assumption proves particularly fragile when dealing with complex physical systems. In the physical sciences, true prediction quality is not defined by average pixel-wise errors but is instead measured by domain-specific metrics that possess clear physical meaning yet are often ***non-differentiable***. These metrics include the Critical Success Index (CSI) (Rasp et al., 2020; Schaefer, 1990; Shu et al., 2025) for evaluating extreme weather events, the Turbulent Kinetic Energy (TKE) spectrum for verifying the physical consistency of fluid systems (Wu et al., 2024d; Wang et al., 2020), or energy norms that ensure adherence to fundamental conservation laws (Müller, 2023). Consequently, a **Fundamental Disconnect** exists between the optimization objectives and the evaluation standards in the current paradigm. *This disconnect leads to models that, even with excellent performance on proxy losses, often fail to capture extreme events critical for scientific decision-making or to maintain physical consistency.* This issue now stands as a core bottleneck hindering the full potential of AI for Science.

To fundamentally address this challenge, we advocate for a **Paradigm Shift**: ***Reframing Forecasting as Planning***. In this new paradigm, we move beyond the goal of passively fitting data. Instead, we treat the forecasting model as an active ***agent*** (Buşoniu et al., 2010) that learns a ***policy*** (Fernández & Veloso, 2006) to make "decisions" - that is, to generate an initial intention or ***action*** (Foerster et al., 2019). This action subsequently guides a learned ***world model*** (Allen & Koomen, 1983) to perform forward-looking exploration, systematically searching through thousands of "imagined" futures to identify those states that maximize a non-differentiable ***reward***. This entire concept finds its most natural theoretical grounding in **Model-Based Reinforcement Learning (MBRL)** (Moerland et al., 2023; Luo et al., 2024), which provides a principled pathway for directly optimizing the domain-specific objectives that truly matter.

Building on this new paradigm, we design and implement a novel framework named SFP, as shown in Figure 1. The core of SFP lies in its two synergistic components. ✎ First, we construct a **Generative World Model** that efficiently learns the complex, stochastic dynamics of the physical system by combining a deterministic base network with a probabilistic Vector Quantization (VQ) (Van Den Oord et al., 2017) module. This model not only predicts a single future but, more critically, generates a diverse and high-fidelity set of future possibilities in "imagination," conditioned on the agent's intention. ✎ Second, we introduce a novel **planning algorithm** that performs efficient exploration among the numerous future trajectories generated by the world model using Beam Search. It directly employs the non-differentiable domain metrics as a reward function to evaluate the quality of each trajectory. Finally, through

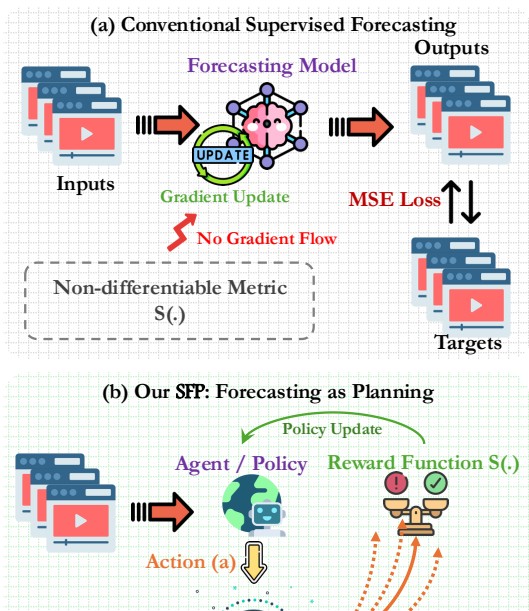

Figure 1: **The SFP Paradigm:** *From Supervised Learning to Planning.* **(a)** The conventional paradigm relies on differentiable proxy losses (e.g., MSE) and fails to incorporate non-differentiable metrics $S(\cdot)$ into the optimization loop. **(b)** Our SFP framework treats forecasting as planning. An *Agent* guides a *Generative World Model* to explore imagined futures. The non-differentiable metric $S(\cdot)$ becomes the *Reward Function*, providing a direct learning signal for the *Policy Update*. This closed-loop process allows the agent to optimize directly for the true objectives of the task.

an iterative **self-training** loop, the high-reward future states discovered via planning are used as high-quality pseudo-labels, which in turn guide the optimization and evolution of the agent's policy, thereby forming the closed-loop learning system illustrated in Figure 1(b). Our contributions can be summarized as follows:

❶ *A New Paradigm*. We are the first to propose reframing spatiotemporal forecasting as a planning task (*Spatiotemporal Forecasting as Planning*) and to systematically formalize it as an MBRL problem. This theoretical framework provides a novel and principled pathway for directly optimizing domain-specific metrics that are critical for scientific discovery but are non-differentiable.

❷ *A Novel Implementation Framework*: SFP. We design and implement SFP, a novel framework that materializes our proposed paradigm. It creatively combines a **Generative World Model**, for exploring diverse future possibilities, with a **Beam Search-based planning algorithm**, for learning from non-differentiable rewards, to form a complete, end-to-end policy learning loop.

❸ *Superior Experimental Performance*. We conduct extensive experiments on several challenging spatiotemporal forecasting benchmarks. The results demonstrate that our framework not only significantly outperforms state-of-the-art methods on traditional accuracy metrics, but more importantly, shows exceptional capabilities in improving physical consistency and capturing extreme events. This provides strong evidence for the effectiveness and superiority of our new paradigm.

## 2 RELATED WORK

**Data-Driven Spatiotemporal Forecasting.** Deep learning models, from CNNs like SimVP (Gao et al., 2022a) to Transformers like FourCastNet (Pathak et al., 2022) and Neural Operators like FNO (Li et al., 2020), have excelled at learning complex dynamics from data. Physics-Informed Neural Networks (PINNs) (Raissi et al., 2019) further improve physical consistency by adding PDE constraints to the loss. However, a fundamental limitation unites them: their reliance on fully differentiable loss functions (e.g., MSE) prevents direct optimization for critical, **non-differentiable** domain metrics like the Critical Success Index (CSI). SFP does not replace these backbones; instead, *it introduces a new, orthogonal optimization paradigm that enables any model to learn directly from these true real-world objectives*.

**Model-Based Reinforcement Learning (MBRL).** MBRL enables efficient, forward-looking decision-making by learning a *world model* of the environment and planning within it (Hafner et al., 2019; 2025). Inspired by this, SFP is the first to systematically apply the MBRL paradigm to physical spatiotemporal forecasting. Our key challenge and contribution lie in adapting this framework to a novel setting: instead of learning from simple, scalar rewards via direct interaction, our agent learns from an *external, high-dimensional, and non-differentiable evaluation function*. This necessitates a novel planning algorithm capable of leveraging such complex reward signals to guide policy learning.

**Generative Forecasting and Complex Rewards.** While generative models like GANs (Goodfellow et al., 2020) and Diffusion Models (Ho et al., 2020) excel at producing diverse forecasts, they typically optimize for data likelihood rather than specific downstream metrics. Concurrently, learning from complex rewards, exemplified by Reinforcement Learning from Human Feedback (RLHF) (Ouyang et al., 2022), has been highly successful in aligning large language models. SFP elegantly unifies these concepts: its generative world model **explores** diverse futures, while its planning mechanism **exploits** this exploration by learning from complex rewards. We frame this as **RL from Metric Feedback (RLMF)**, extending RLHF from human preferences to any computable domain metric. Unlike traditional self-training based on model confidence, RLMF derives its learning signal from an external evaluation of exploration outcomes, making the process more targeted and powerful.

## 3 PROBLEM FORMULATION: REFRAMING FORECASTING AS PLANNING

Conventional spatiotemporal forecasting aims to learn a mapping $f_\theta : \mathcal{X}_t \mapsto \hat{\mathbf{y}}_{t+1}$, where the parameters $\theta$ are optimized by minimizing a differentiable proxy loss, such as the MSE. However, this paradigm cannot directly optimize the **non-differentiable domain metrics**, $\mathcal{S}(\cdot)$, such as the CSI, which are critical for evaluating performance in the physical sciences.

To address this fundamental disconnect, we propose to reframe spatiotemporal forecasting as planning and formalize it as a Model-Based Reinforcement Learning problem. In this paradigm, we do not directly predict $\mathbf{y}_{t+1}$. Instead, we learn a **policy** $\pi_\theta$ that, given the current state $\mathbf{s}_t = \mathcal{X}_t$, generates a high-dimensional continuous **action** $\mathbf{a}_t = \pi_\theta(\mathbf{s}_t)$. This action represents an initial predictive intention that guides a learned *Generative World Model*, $\mathcal{M}_\phi$, to perform forward exploration. The

world model defines the environment's transition dynamics $p_\phi(\mathbf{y}_{t+1}|\mathbf{a}_t)$, which we approximate by sampling a set of $K$ candidate states $\{\tilde{\mathbf{y}}_{t+1}^{(k)}\}_{k=1}^{K}$.

Our core idea is to employ the non-differentiable metric $\mathcal{S}(\cdot)$ as a **reward function**, $\mathcal{R}$, which evaluates the future states explored within the world model, rather than the initial action. Our ultimate goal is thus to learn an optimal policy $\pi_\theta^*$ that maximizes the expected return defined by this reward function. We express this learning objective as:

$$\pi_\theta^* = \arg\max_{\pi_\theta} \mathbb{E}_{\mathbf{s}_t \sim \mathcal{D}}\left[\mathcal{R}(\pi_\theta(\mathbf{s}_t))\right], \quad \text{where} \quad \mathcal{R}(\mathbf{a}_t) = \mathbb{E}_{\tilde{\mathbf{y}} \sim p_\phi(\cdot|\mathbf{a}_t)}[\mathcal{S}(\tilde{\mathbf{y}})] \tag{1}$$

Equation equation 1 forms the theoretical cornerstone of our `SFP` framework. Since the reward function $\mathcal{R}$ depends on a complex, non-differentiable generation and evaluation process, optimizing $\pi_\theta$ via direct backpropagation is infeasible. The following sections detail how we address this optimization challenge by jointly learning the world model $\mathcal{M}_\phi$ and using a novel planning algorithm.

## 4 THE SFP FRAMEWORK

Our `SFP` framework operationalizes the ***Spatiotemporal Forecasting as Planning*** paradigm through a decoupled, two-stage process designed to solve the optimization objective in Equation equation 1. ❶. First, we pre-train a generative world model, $\mathcal{M}_\phi$, to learn the system's probabilistic dynamics and provide a high-fidelity "imagination" space. ❷. Subsequently, with the world model's parameters frozen, we optimize the policy, $\pi_\theta$, in an iterative loop. In each iteration, the policy's action guides the world model's exploration of future states; a non-differentiable reward function assesses these outcomes; and the highest-reward state is then used as a pseudo-label to update the policy via self-training. This separation of world model learning from planning-based policy optimization allows `SFP` to effectively translate non-differentiable reward signals into feasible gradient updates, systematically solving our formulated objective.

### 4.1 STAGE 1: LEARNING THE GENERATIVE WORLD MODEL

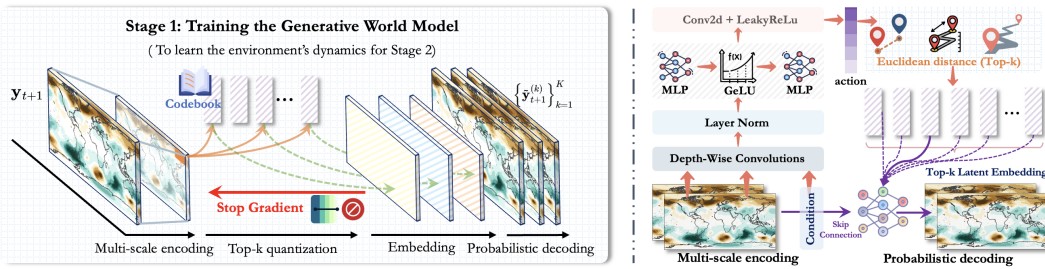

Figure 2: **Architecture of our Generative World Model** ($\mathcal{M}_\phi$). Operating as a conditional VQ-VAE, its probabilistic decoder fuses a latent **action** embedding with a **condition** embedding derived from the current state $\mathbf{s}_t$. This design enables the generation of a distribution of $K$ diverse future states based on the agent's intention.

The quality of the world model, $\mathcal{M}_\phi$, directly dictates the upper bound of the subsequent planning stage (Stage 2), as it constitutes the virtual "imagination" space for the agent. Therefore, in this first stage, our core task is to learn a high-fidelity model that accurately captures the conditional probability dynamics of the physical system, $p_\phi(\mathbf{y}_{t+1}|\mathbf{s}_t)$. To this end, we design an architecture based on a Conditional Vector-Quantized Variational Autoencoder (Conditional VQ-VAE), as illustrated in Figure 2. Our world model $\mathcal{M}_\phi$ consists of a multi-scale encoder $E_\phi$ and a conditional decoder $D_\phi$. Given a ground-truth future state $\mathbf{y}_{t+1}$ from the dataset, the encoder $E_\phi$ first maps it to a continuous latent feature map $\mathbf{z}_e(\mathbf{y}_{t+1}) \in \mathbb{R}^{h \times w \times d}$ through a series of multi-scale convolutions.

Next, we introduce a learnable discrete codebook $\mathcal{C} = \{\mathbf{e}_i\}_{i=1}^{N} \subset \mathbb{R}^d$, where $N$ is the number of code vectors and $d$ is their dimensionality. Through a vector quantization process, we deterministically replace each vector in the continuous latent map $\mathbf{z}_e$ with its nearest neighbor from the codebook $\mathcal{C}$ in terms of Euclidean distance. This process yields a discretized latent representation $\mathbf{z}_q(\mathbf{y}_{t+1})$:

$$\mathbf{z}_q(\mathbf{y})_{i,j} = \mathbf{e}_{k^*}, \quad \text{where} \quad k^* = \arg\min_{k \in \{1,\dots,N\}} \|\mathbf{z}_e(\mathbf{y})_{i,j} - \mathbf{e}_k\|_2^2 \tag{2}$$

Figure 3: *The architecture of Stage 2: Iterative Policy Optimization via Planning and Self-Training.* The process unfolds in a closed loop. **(1)** *Agent Decides*: Given the current state $s_t$, the trainable Agent (Policy $\pi_\theta$, marked by a 🔥) generates a latent action $a_t$. **(2)** *World Model Imagines*: The frozen Generative World Model ($M_\phi$, marked by a ❄) uses this action to perform forward exploration within its "Imagination Space," producing a distribution of diverse future states $\{\hat{y}_{t+1}^{(k)}\}$. **(3)** *Planner Evaluates*: A planning algorithm leverages a non-differentiable domain metric as a Reward Function to identify the highest-reward future, $\hat{y}_{t+1}^*$. **(4)** *Policy Self-Updates*: This high-reward future serves as a high-quality pseudo-label to update the agent's policy $\pi_\theta$ via a standard differentiable loss.

Unlike a standard VQ-VAE, our decoder $D_\phi$ is **conditional**. As shown in Figure 2 (right), it receives not only the quantized latent representation $\mathbf{z}_q$ from the future state but also a **condition** vector $\mathbf{c}_t = C_\phi(\mathbf{s}_t)$ extracted from the current state $\mathbf{s}_t = \mathcal{X}_t$. The decoder's task is to fuse these two information sources to reconstruct the original future state, i.e., $\tilde{\mathbf{y}}_{t+1} = D_\phi(\mathbf{z}_q, \mathbf{c}_t)$. This conditional mechanism is crucial as it ensures that the futures generated by the world model evolve in a manner consistent with the historical context.

**Training Objective.** We optimize the parameters $\phi$ of the entire world model end-to-end using a composite loss function, $\mathcal{L}_{\text{WM}}$. This loss comprises three components designed to minimize reconstruction error while regularizing the behavior of the encoder and the codebook:

$$\mathcal{L}_{\text{WM}}(\phi) = \underbrace{\|\mathbf{y}_{t+1} - D_\phi(\mathbf{z}_q(\mathbf{y}_{t+1}), \mathbf{c}_t)\|_2^2}_{\text{Reconstruction Loss}} + \underbrace{\|\text{sg}[\mathbf{z}_e(\mathbf{y}_{t+1})] - \mathbf{z}_q(\mathbf{y}_{t+1})\|_2^2}_{\text{Codebook Loss}} + \underbrace{\beta\|\mathbf{z}_e(\mathbf{y}_{t+1}) - \text{sg}[\mathbf{z}_q(\mathbf{y}_{t+1})]\|_2^2}_{\text{Commitment Loss}}$$

(3)

Here, sg[·] denotes the stop-gradient operator. The reconstruction loss optimizes the encoder and decoder; the codebook loss updates the codebook by pulling the selected code vectors towards the encoder's outputs; and the commitment loss encourages the encoder's output to remain close to the chosen code vector, which stabilizes the training process. The hyperparameter $\beta$ balances the contribution of the commitment loss.

By minimizing $\mathcal{L}_{\text{WM}}$ on real data pairs $(\mathbf{s}_t, \mathbf{y}_{t+1})$, we obtain a high-quality generative world model $\mathcal{M}_\phi$. Upon entering the next stage, the parameters $\phi$ of this model are **frozen**, allowing it to serve as a stable and reliable simulation environment for policy optimization.

### 4.2 STAGE 2: POLICY OPTIMIZATION VIA PLANNING AND SELF-TRAINING

With a high-fidelity, pre-trained generative world model $\mathcal{M}_\phi$ at our disposal, we now address the core challenge of optimizing the policy $\pi_\theta$. As established, the non-differentiable nature of the reward function $\mathcal{R}$ in Equation equation 1 precludes direct optimization via backpropagation. Stage 2 introduces a novel iterative loop that translates these black-box reward signals into tractable, differentiable supervision for the policy. This process, illustrated in Figure 3, hinges on a synergistic interplay of planning, evaluation, and self-training.

First, we define our **agent** as the predictive model governed by the policy $\pi_\theta$. The agent's **action**, $\mathbf{a}_t = \pi_\theta(\mathbf{s}_t)$, is not a direct prediction but rather a high-dimensional, continuous latent vector. This vector serves as a *latent directive* that steers the generative process of the world model, encoding the agent's initial intention for the future state.

The cornerstone of this stage is a **planning algorithm** that performs lookahead inference within the learned world model. Given an action $\mathbf{a}_t$, which defines the initial condition for exploration, we employ **Beam Search** to efficiently navigate the vast "Imagination Space" of possible futures. The algorithm maintains a "beam" of $B$ most promising partial sequences at each step, progressively expanding them to generate a set of $B$ high-quality, full-length future state candidates, denoted as $\{\hat{\mathbf{y}}_{t+1}^{(b)}\}_{b=1}^{B}$. This forward exploration is computationally inexpensive as it occurs entirely within the frozen world model $\mathcal{M}_\phi$.

The climax of the loop is the **reward evaluation and self-training** mechanism. Each candidate future $\hat{\mathbf{y}}_{t+1}^{(b)}$ generated by the planner is evaluated using the domain-specific, non-differentiable metric $\mathcal{S}(\cdot)$ as the reward function. The future state that yields the maximum reward is identified as the optimal outcome of the exploration:

$$\hat{\mathbf{y}}_{t+1}^{*} = \underset{\hat{\mathbf{y}}_{t+1}^{(b)}}{\arg\max} \left[ \mathcal{S}\left(\hat{\mathbf{y}}_{t+1}^{(b)}\right) \right], \quad \text{for } b \in \{1, \dots, B\} \tag{4}$$

This highest-reward state, $\hat{\mathbf{y}}_{t+1}^{*}$, discovered through planning, serves as a high-quality **pseudo-label**. It represents a desirable future that the agent should aim to produce. Consequently, we formulate a differentiable **policy loss** to minimize the discrepancy between a projection of the agent's action and this pseudo-label. While various forms are possible, a common objective is the Mean Squared Error:

$$\mathcal{L}_{\text{policy}}(\theta) = \mathbb{E}_{(\mathbf{s}_t, \hat{\mathbf{y}}_{t+1}^{*})} \left[ \|\mathcal{P}(\pi_\theta(\mathbf{s}_t)) - \hat{\mathbf{y}}_{t+1}^{*}\|_2^2 \right] \tag{5}$$

Here, $\mathcal{P}(\cdot)$ is a simple, lightweight projector that maps the latent action $\mathbf{a}_t$ back to the physical state space. Since $\mathcal{L}_{\text{policy}}$ is fully differentiable with respect to the policy parameters $\theta$, we can update the policy via standard gradient descent. This self-training loop effectively distills the knowledge from the non-differentiable reward into the policy network, iteratively refining the agent's ability to propose actions that lead to high-reward futures.

## 5 EXPERIMENTS

We conduct a series of comprehensive experiments to validate the effectiveness and superiority of our proposed SFP. Our experiments are designed to answer the following key research questions (RQs): *RQ1*: **General Efficacy**. Does SFP consistently outperform conventional supervised training, especially on non-differentiable domain-specific metrics? *RQ2*: **Data Scarcity Mitigation**. Does SFP's performance advantage grow as the amount of training data decreases? *RQ3*: **Cost-Benefit Analysis**. Are SFP's performance gains worth the additional computational overhead in time and memory? *RQ4*: **Ablation Study**. What are the contributions of SFP's core components (planning, self-training, rewards) and its sensitivity to key hyperparameters? *RQ5*: **Training Stability and Probabilistic Skill**: How does SFP compare to methods like Direct Preference Optimization (DPO) in terms of training stability? Can it generate high-quality probabilistic ensembles?

**Experimental Settings.** Our experimental validation is conducted on five diverse benchmarks to ensure robust and generalizable conclusions. These include real-world datasets like *SEVIR* Veillette et al. (2020) for extreme weather and a *Marine Heatwave* dataset for data-scarce scenarios, as well as classic equation-driven systems from PDEBench Takamoto et al. (2022) (**NSE**, **SWE**, **RBC**) and the high-fidelity CFD simulation *Prometheus* Wu et al. (2024b). We employ a wide range of backbone models to demonstrate the plug-and-play nature of SFP, spanning conventional architectures (*SimVP-v2* Tan et al. (2022), *ConvLSTM* Shi et al. (2015), *Earthformer* Gao et al. (2022b)) and Neural Operators (*FNO* Li et al. (2020), *CNO* Raonic et al. (2023)).

Our evaluation protocol is multi-faceted. Beyond standard accuracy metrics like **MSE** and **RMSE**, we focus on critical, non-differentiable domain metrics that also serve as reward signals: the **Critical Success Index (CSI)** for extreme events, the *Turbulent Kinetic Energy (TKE)* spectrum for physical consistency, and the *Structural Similarity Index (SSIM)*. For probabilistic evaluation, we report the *Continuous Ranked Probability Score (CRPS)*. All experiments are implemented in PyTorch on NVIDIA A100 GPUs. We use the Adam optimizer with a learning rate of $1 \times 10^{-3}$ and a cosine annealing schedule. Key SFP hyperparameters are a 1024-entry codebook and a beam width $B = 10$ for planning, with a detailed sensitivity analysis in Section 5.

Table 1: Performance comparison of SFP against supervised training on three representative benchmarks. SFP shows substantial gains on critical, non-differentiable domain metrics (CSI, TKE Error, and SSIM) while also improving standard accuracy (MSE). Lower is better (↓) for MSE and TKE Error; higher is better (↑) for CSI and SSIM. Best results are in **bold**.

| Model | SEVIR (Extreme Weather) | | | | NSE (Turbulence) | | | | Prometheus (Combustion) | | | |
|---|---|---|---|---|---|---|---|---|---|---|---|---|
| | MSE ↓ | | CSI ↑ | | MSE ↓ | | TKE Error ↓ | | MSE ↓ | | SSIM ↑ | |
| | Baseline | + SFP | Baseline | + SFP | Baseline | + SFP | Baseline | + SFP | Baseline | + SFP | Baseline | + SFP |
| ResNet | 0.0671 | **0.0542** | 0.32 | **0.45** | 0.2330 | **0.1663** | 0.48 | **0.25** | 0.2356 | **0.1987** | 0.72 | **0.81** |
| ConvLSTM | 0.1757 | **0.1283** | 0.28 | **0.42** | 0.4094 | **0.1277** | 0.55 | **0.18** | 0.0732 | **0.0533** | 0.88 | **0.94** |
| Earthformer | 0.0982 | **0.0521** | 0.48 | **0.62** | 1.8720 | **0.1202** | 0.68 | **0.15** | 0.2765 | **0.2001** | 0.79 | **0.86** |
| SimVP-v2 | 0.0063 | **0.0032** | 0.52 | **0.65** | 0.1238 | **0.1022** | 0.39 | **0.16** | 0.1238 | **0.0921** | 0.85 | **0.92** |
| TAU | 0.0059 | **0.0029** | 0.54 | **0.68** | 0.1205 | **0.1017** | 0.40 | **0.17** | 0.1201 | **0.0899** | 0.86 | **0.93** |
| Earthfarseer | 0.0065 | **0.0021** | 0.55 | **0.70** | 0.1138 | **0.0987** | 0.38 | **0.19** | 0.1176 | **0.1092** | 0.87 | **0.91** |
| FNO | 0.0783 | **0.0436** | 0.35 | **0.51** | 0.2237 | **0.1005** | 0.41 | **0.17** | 0.3472 | **0.2275** | 0.75 | **0.84** |
| NMO | 0.0045 | **0.0029** | 0.58 | **0.72** | 0.1007 | **0.0886** | 0.35 | **0.15** | 0.0982 | **0.0475** | 0.89 | **0.95** |
| CNO | 0.0056 | **0.0053** | 0.56 | **0.64** | 0.2188 | **0.1483** | 0.45 | **0.22** | 0.1097 | **0.0254** | 0.84 | **0.96** |
| FourcastNet | 0.0721 | **0.0652** | 0.51 | **0.60** | 0.1794 | **0.1424** | 0.32 | **0.21** | 0.0987 | **0.0542** | 0.90 | **0.95** |
| Avg. Improv. (%) | -34.9% | | +29.7% | | -56.2% | | -57.3% | | -33.6% | | +11.5% | |

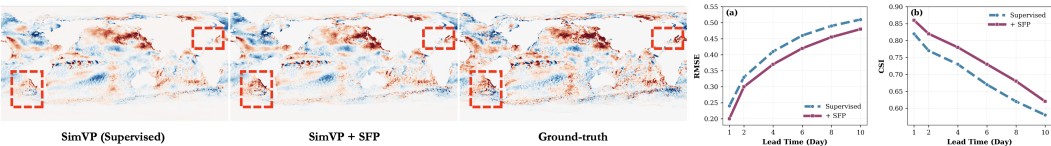

Figure 4: SFP **excels at capturing extreme marine heatwaves. (Left)** On day 10, SFP successfully predicts critical heatwave regions (red boxes) missed by the supervised baseline. **(Right)** Quantitative curves show SFP's consistent lead in RMSE and a significantly superior CSI, highlighting its skill in forecasting rare events.

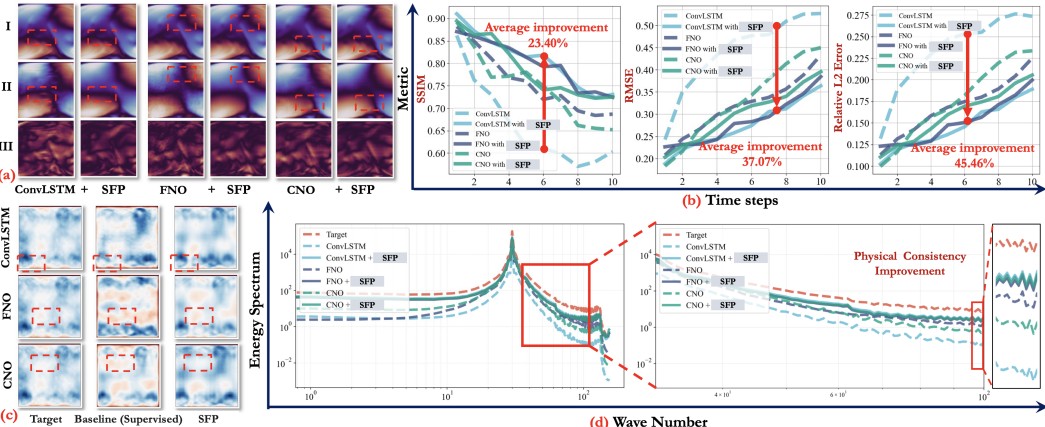

Figure 5: SFP **enhances physical consistency in turbulence forecasting (NSE dataset). (a-c)** SFP shows sustained improvements on standard metrics and generates more realistic, fine-grained vortex structures. **(d)** The energy spectrum analysis confirms SFP's physical fidelity: its spectrum (solid lines) closely matches the ground truth, especially in the high-frequency regime, unlike baselines (dashed lines) which exhibit severe energy distortion.

**General Efficacy of the SFP Paradigm (*RQ1*)** We first evaluate the general efficacy of SFP by applying it across ten backbone models on several challenging benchmarks. Our results show that SFP not only consistently improves standard accuracy metrics but, more critically, delivers breakthrough performance on non-differentiable, domain-specific metrics crucial for real-world applications and physical fidelity.

*Quantitative Analysis.* Table 1 demonstrates SFP's remarkable versatility, outperforming supervised baselines across all tested scenarios. The advantage is most pronounced on domain-specific metrics. For instance, on the SEVIR dataset, SFP boosts the average CSI by a substantial **29.7%**, critical for extreme weather nowcasting. On the highly challenging NSE turbulence task, SFP slashes the TKE spectrum error by an average of **57.3%**, highlighting its exceptional ability to preserve physical

consistency. These findings confirm that SFP effectively bridges the gap between conventional training objectives and true evaluation criteria by optimizing for domain-specific reward signals.

*Extreme Event Capturing.* Figure 4 visually confirms SFP's superiority in forecasting rare but critical marine heatwaves. While the supervised baseline fails to predict key high-intensity cores on day 10, SFP accurately captures their intensity and spatial extent. The quantitative curves further show that SFP's lead in CSI grows over the forecast horizon, showcasing our planning-based paradigm's strength in discovering and generating high-reward, low-probability future states.

*Physical Consistency.* Figure 5 provides deeper insights into SFP's physical fidelity. In the NSE turbulence task, vorticity visualizations (panel c) reveal that SFP generates richer, more realistic fine-grained vortex structures. This is decisively quantified by the energy spectrum analysis (panel d), where the spectra of SFP-enhanced models (solid lines) closely match the ground truth, especially in the high-frequency regime. In contrast, baselines (dashed lines) exhibit severe energy distortion, a common failure of MSE-based optimization. SFP overcomes this by planning for physically plausible trajectories within its learned world model.

**Robustness to Data Scarcity (RQ2)** To validate the robustness of SFP in data-scarce regimes, we conduct comparative experiments on training subsets of varying sizes (10%-100%). As illustrated in Figure 6, the performance advantage of SFP over the supervised baseline becomes more pronounced as the amount of training data decreases. The gap between the two curves is widest at the lowest data ratio, a trend that is particularly evident

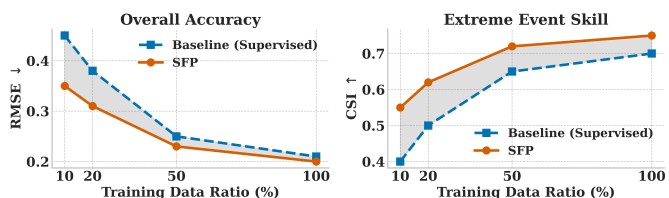

Figure 6: SFP **demonstrates strong robustness in data-scarce regimes.** Performance gap between SFP (red) and the supervised baseline (blue) widens as training data decreases, especially on the critical CSI metric for extreme events.

on the critical CSI metric for extreme event forecasting. This result strongly demonstrates that SFP's planning and self-training mechanism effectively generates high-quality pseudo-labels to compensate for the lack of real data, thereby significantly enhancing the model's generalization and performance in low-data settings.

**Cost-Benefit Analysis (RQ3).** Although SFP delivers significant performance improvements, it also introduces additional computational complexity. To provide a clear cost-benefit analysis, we evaluate the computational overhead of SFP compared to the supervised baseline across various configurations, with results summarized in Table 2. The data reveals that SFP increases training time and model parameters by a moderate margin, typically around $1.4\times$, due to the inclusion of the generative world model. The most notable overhead is inference latency, which increases significantly due to the beam search planning process. However, this computational investment yields disproportionately large returns in predictive accuracy and physical fidelity. For example, on the challenging NSE task, SFP achieves a remarkable **-77.9%** reduction in TKE error on the Earthformer backbone for a ~$1.4\times$ increase in training time. In critical scientific applications where predictive skill is paramount, this trade-off is highly favorable, establishing SFP as a practical and valuable paradigm.

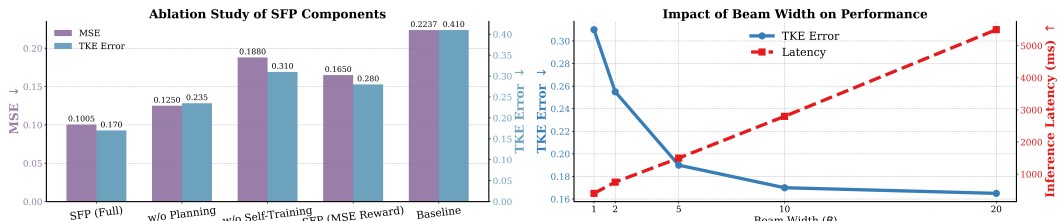

Figure 7: **Ablation and sensitivity analysis of** SFP**. (Left)** Each component is crucial, especially the non-differentiable TKE reward. **(Right)** A beam width of $B = 10$ provides a strong balance between performance and latency.

Table 2: **Cost-benefit analysis of** SFP. SFP yields substantial performance gains for a moderate increase in computational overhead. Overhead values are presented as *Baseline* / SFP. Key Metric refers to CSI for SEVIR, TKE Error for NSE, and SSIM for Prometheus.

| Benchmark | Model | Performance Gain | | Training Overhead | | | Inference Overhead | |
|---|---|---|---|---|---|---|---|---|
| | | MSE (%) ↓ | Key Metric (%) ↑↓ | Time (h) | Mem (GB) | Params (M) | Latency (ms) | Mem (GB) |
| SEVIR | ConvLSTM | -27.0 | +42.8 | 8.5 / 12.0 | 12.1 / 13.5 | 25.8 / 42.1 | 210 / 1650 | 6.5 / 8.2 |
| | SimVP-v2 | -49.2 | +25.0 | 5.2 / 7.5 | 10.5 / 11.8 | 18.5 / 33.7 | 150 / 1250 | 5.8 / 7.1 |
| | FNO | -44.3 | +45.7 | 12.5 / 18.0 | 15.2 / 16.5 | 45.1 / 65.3 | 250 / 1800 | 8.1 / 10.3 |
| NSE | ConvLSTM | -68.8 | -67.3 | 15.0 / 22.5 | 18.5 / 20.1 | 30.2 / 50.5 | 450 / 3800 | 10.2 / 12.8 |
| | Earthformer | -93.6 | -77.9 | 35.2 / 50.1 | 28.9 / 32.5 | 95.7 / 125.1 | 880 / 7500 | 18.5 / 22.4 |
| | FNO | -55.1 | -58.5 | 20.5 / 29.0 | 20.1 / 22.3 | 50.8 / 72.4 | 520 / 4100 | 11.5 / 14.1 |
| Prometheus | SimVP-v2 | -25.8 | +8.2 | 10.1 / 14.5 | 16.2 / 17.8 | 22.1 / 38.9 | 330 / 2850 | 9.1 / 11.5 |
| | Earthformer | -27.6 | +8.9 | 40.8 / 58.0 | 30.5 / 34.1 | 102.3 / 133.7 | 1100 / 9200 | 20.2 / 24.8 |
| | CNO | -76.8 | +14.3 | 25.6 / 36.2 | 22.8 / 25.0 | 60.5 / 85.1 | 650 / 5500 | 13.8 / 16.9 |

Figure 8: SFP**'s advanced capabilities in stability and probabilistic forecasting (***RQ5***). (Left)** Across diverse backbones, Direct DPO suffers from training collapse while SFP maintains stable convergence. **(Right)** SFP achieves state-of-the-art probabilistic skill, outperforming the deterministic baseline and the specialized generative model, PreDiff, on the CRPS metric.

**Ablation and Sensitivity Analysis (*RQ4*).** Our ablation studies, shown in Figure 7, confirm the importance of each component within the SFP framework. The bar chart (left) shows that removing any key element degrades performance, with the most significant drop in physical consistency (TKE Error) occurring when the domain-specific TKE reward is replaced by a standard MSE. This highlights that optimizing directly for relevant, nondifferentiable metrics is the cornerstone of SFP's success. Furthermore, the sensitivity analysis (right) reveals a clear trade-off between performance and latency as a function of beam width $B$. The results justify our choice of $B = 10$ as a default, as it achieves near-optimal accuracy at a manageable computational cost, demonstrating the practical robustness of the framework.

**Advanced Capabilities: Stability & Probabilistic Skill (*RQ5*).** We conclude our analysis by evaluating SFP's advanced capabilities, focusing on training stability and probabilistic skill. Figure 8 summarizes the key findings. The training curves (left) reveal a critical advantage: while Direct DPO frequently collapses during training across different backbones, SFP consistently and stably converges. This superior stability stems from our decoupled world model, which provides a robust foundation for planning-based exploration. Furthermore, the bar charts (right) demonstrate SFP's state-of-the-art probabilistic forecasting ability. By forming an ensemble from its planned trajectories, SFP achieves the best (lowest) CRPS score on both benchmarks, outperforming even specialized generative models like PreDiff. These results establish SFP as a robust, high-performance paradigm for both deterministic and probabilistic forecasting.

## 6 CONCLUSION

In this work, we introduce a new paradigm, spatial-temporal forecasting as planning, reframing the traditional forecasting task as a model-based reinforcement learning problem. Our proposed framework, SFP, effectively addresses the long-standing challenge of optimizing for nondifferentiable, domain-specific metrics by integrating a generative world model with a planning-based policy optimization process. Comprehensive experiments demonstrate that SFP not only significantly improves predictive accuracy across a wide range of backbones but also excels at capturing extreme events, maintaining physical consistency, and generating high-quality probabilistic forecasts, all while ensuring robust training stability. By bridging the gap between differentiable proxy losses and true scientific objectives, SFP paves the way for developing more reliable and impactful AI for science.

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

# A   THE USE OF LARGE LANGUAGE MODELS (LLMS)

LLMs were not involved in the research ideation or the writing of this paper.

# B   METRIC

## B.1   STANDARD EVALUATION METRICS

**Mean Squared Error (MSE)** MSE is a common statistical metric used to assess the difference between predicted and actual values. The formula is:

$$\text{MSE} = \frac{1}{n} \sum_{i=1}^{n} (y_i - \hat{y}_i)^2 \tag{6}$$

where $n$ is the number of samples, $y_i$ is the actual value, and $\hat{y}_i$ is the predicted value.

**Relative L2 Error** Relative L2 error measures the relative difference between predicted and actual values, commonly used in time series prediction. The formula is:

$$\text{Relative L2 Error} = \frac{\|Y_{\text{pred}} - Y_{\text{true}}\|_2}{\|Y_{\text{true}}\|_2} \tag{7}$$

where $Y_{\text{pred}}$ is the predicted value and $Y_{\text{true}}$ is the actual value.

**Continuous Ranked Probability Score (CRPS)** The Continuous Ranked Probability Score (CRPS) is a proper scoring rule used to assess the quality of probabilistic forecasts. It generalizes the Mean Absolute Error (MAE) to probabilistic forecasts; for a deterministic forecast, CRPS reduces to MAE. The CRPS is defined by the integral of the squared difference between the forecast Cumulative Distribution Function (CDF) $F$ and the empirical CDF of the observation $y$:

$$\text{CRPS}(F, y) = \int_{-\infty}^{\infty} (F(x) - H(x - y))^2 \, dx \tag{8}$$

where $H(x - y)$ is the Heaviside step function, which is 0 for $x < y$ and 1 for $x \geq y$. For an ensemble forecast with $M$ members $\{x_i\}_{i=1}^{M}$, CRPS can be more intuitively computed as:

$$\text{CRPS}(\{x_i\}, y) = \frac{1}{M} \sum_{i=1}^{M} |x_i - y| - \frac{1}{2M^2} \sum_{i=1}^{M} \sum_{j=1}^{M} |x_i - x_j| \tag{9}$$

This formulation highlights that CRPS rewards accuracy (the first term, average absolute error of ensemble members) and sharpness (the second term, a penalty for large spread among members). Lower CRPS values indicate a better forecast.

## B.2   DETAILED DEFINITIONS OF DOMAIN-SPECIFIC REWARD METRICS

To clarify the physical significance of the non-differentiable metrics used as reward functions in our SFP framework, we provide their detailed mathematical definitions below.

**1. Critical Success Index (CSI)**
Used as the reward signal for the SEVIR (Extreme Weather) benchmark. CSI is a standard meteorological metric for evaluating the prediction of rare events (e.g., heavy precipitation). Unlike MSE, it ignores true negatives and focuses on the "hit rate" of the target event. Given a binarized prediction map based on a specific threshold $\tau$, the CSI is calculated using the confusion matrix counts:

$$\text{CSI} = \frac{\text{Hits}}{\text{Hits} + \text{Misses} + \text{False Alarms}} \tag{10}$$

where Hits represents correctly predicted event pixels, Misses represents observed events that were not predicted, and False Alarms represents predicted events that did not occur. Optimizing CSI directly allows the model to capture extreme events that are typically smoothed out by standard regression losses.

**2. Turbulent Kinetic Energy (TKE) Error**

Used as the reward signal for the NSE (Fluid Dynamics) benchmark. TKE measures the physical consistency of the flow by evaluating the energy distribution across different spatial scales. We calculate the energy spectrum $E(k)$ of the velocity field $\mathbf{u}$. Let $\hat{\mathbf{u}}(\mathbf{k})$ be the discrete Fourier transform of the velocity field, where $\mathbf{k}$ is the wave vector. The energy at scalar wavenumber $k$ is given by summing over spherical shells:

$$E(k) = \sum_{k-\frac{1}{2} < \|\mathbf{k}\| \leq k+\frac{1}{2}} \|\hat{\mathbf{u}}(\mathbf{k})\|^2 \tag{11}$$

The TKE Error is defined as the distance (e.g., RMSE or Log-Spectral Distance) between the predicted energy spectrum $E_{\text{pred}}(k)$ and the ground truth spectrum $E_{\text{true}}(k)$. Minimizing this error ensures the model adheres to fundamental conservation laws and correctly models the energy cascade, avoiding spectral bias.

**3. Structural Similarity Index (SSIM)**

Used as the reward signal for the Prometheus (Combustion) benchmark. SSIM assesses the structural fidelity of the simulation, which is critical for analyzing complex flame fronts. Given two image windows $x$ and $y$, SSIM combines luminance ($l$), contrast ($c$), and structure ($s$) measurements:

$$\text{SSIM}(x,y) = \frac{(2\mu_x\mu_y + C_1)(2\sigma_{xy} + C_2)}{(\mu_x^2 + \mu_y^2 + C_1)(\sigma_x^2 + \sigma_y^2 + C_2)} \tag{12}$$

where $\mu_x, \mu_y$ are the local means, $\sigma_x^2, \sigma_y^2$ are the local variances, $\sigma_{xy}$ is the covariance, and $C_1, C_2$ are small constants for stability.

## C  MORE EXPERIMENTS

### C.1  INTERPRETATION ANALYSIS

***Qualitative Analysis Using t-SNE.*** Figure 9 shows t-SNE visualizations on the RBC dataset: (a) ground truth, (b) ConvLSTM predictions, and (c) ConvLSTM + `SFP` predictions. In (a), the ground truth has clear clusters. In (b), ConvLSTM's clustering is blurry with overlaps, indicating limited capability in capturing data structure. In (c), ConvLSTM + `SFP` yields clearer clusters closer to the ground truth, demonstrating that `SFP` significantly enhances the model's predictive accuracy and physical consistency.

***Analysis on Code Bank.*** We train FNO+`SFP` on NSE for 100 epochs with a learning rate of 0.001 and batch size of 100. In the VQVAE codebank dimension experiment, increasing the number of vectors $L$ notably reduces MSE. When $L = 1024$ and $D = 64$, the MSE reaches a minimum of 0.1271. Although MSE fluctuates more at $L = 256$ or $512$, overall, higher $L$ helps improve accuracy. Most training losses quickly stabilize within 20 epochs; $L = 512$ and $D = 128$ notably shows higher stability, but $L = 1024$ and $D = 64$ achieves the lowest MSE.

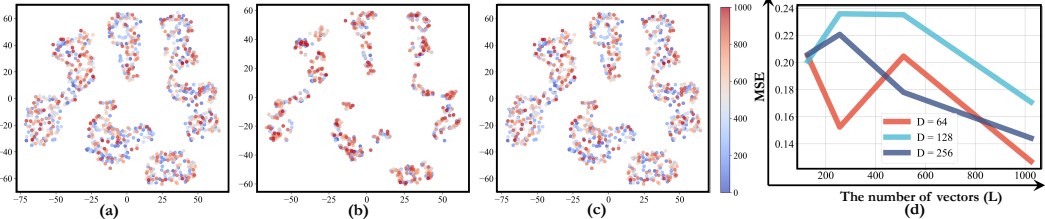

Figure 9: The t-SNE visualization in (a), (b), and (c) shows the Ground-truth, ConvLSTM and ConvLSTM+`SFP` predictions, respectively. (d) shows the analysis of the Codebank parameters.

### C.2  ADDITIONAL EXPERIMENTS

#### C.2.1  LONG-TERM FORECASTING EXPERIMENT EXPANSION

In the long-term forecasting experiments, we compare the performance of different backbone models on the SWE benchmark, evaluating the relative L2 error for three variables (U, V, and H). Our setup

inputs 5 frames and predicts 50 frames. For the SimVP-v2 model, using SFP reduces the relative L2 error for SWE (u) from 0.0187 to 0.0154, SWE (v) from 0.0387 to 0.0342, and SWE (h) from 0.0443 to 0.0397. We visualize SWE (h) in 3D as shown in Figure 10 [I]. For the ConvLSTM model, applying SFP reduces the relative L2 error for SWE (u) from 0.0487 to 0.0321, SWE (v) from 0.0673 to 0.0351, and SWE (h) from 0.0762 to 0.0432. For the FNO model, using SFP reduces the relative L2 error for SWE (u) from 0.0571 to 0.0502, SWE (v) from 0.0832 to 0.0653, and SWE (h) from 0.0981 to 0.0911. Overall, SFP significantly improves the long-term forecasting accuracy of different backbone models.

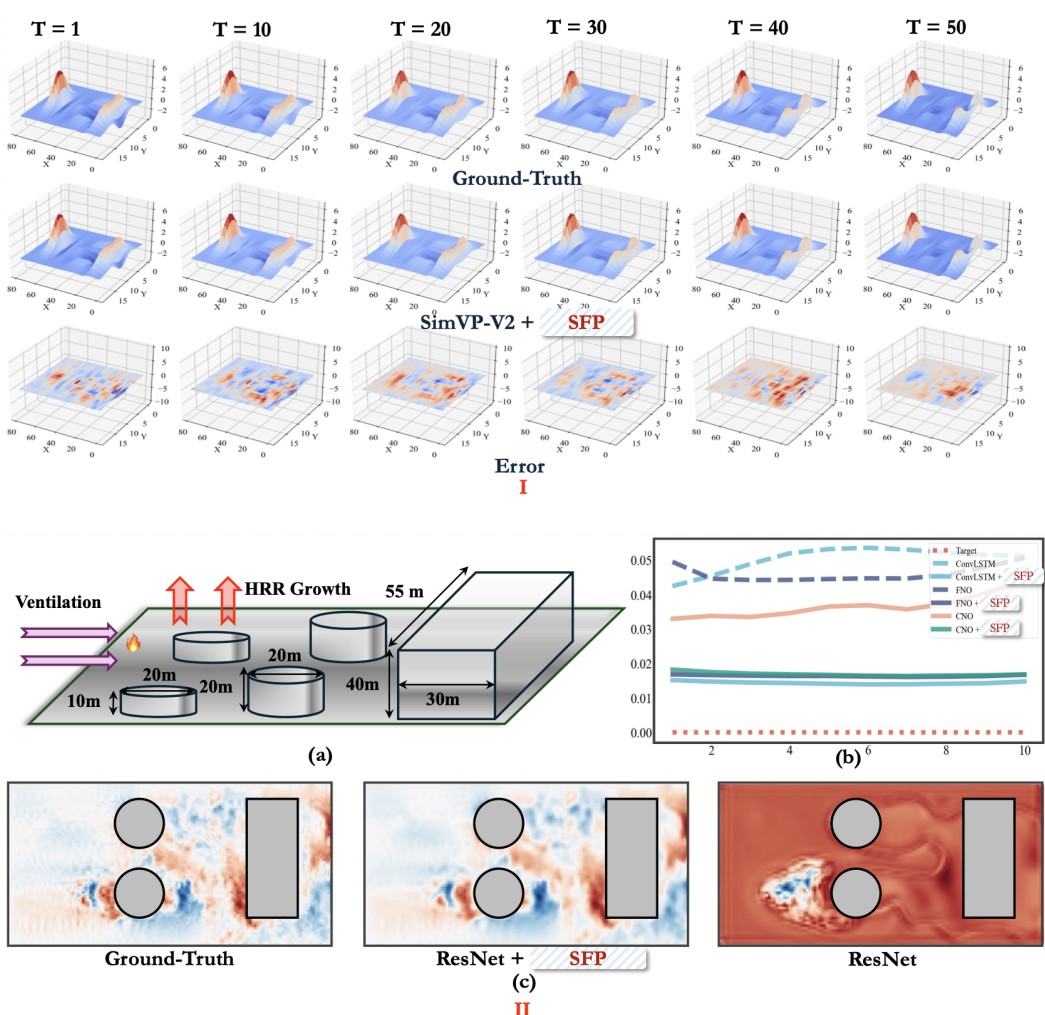

Figure 10: I. 3D visualization of the SWE(h), showing Ground-truth, SimVP-V2+SFP predictions, and Error at T=1, 10, 20, 30, 40, 50. The first row shows Ground-truth, the second SimVP-V2+SFP predictions, and the third Error. II. A case study. Building fire simulation with ventilation settings added to Wu's Prometheus (Wu et al., 2024b). (a) Layout and HRR growth. (b) Comparison of physical metrics for different methods. (c) Ground-truth, ResNet+SFP, and ResNet predictions.

### C.2.2 EXPERIMENT STATISTICAL SIGNIFICANCE

To measure the statistical significance of our main experiment results, we choose three backbones to train on two datasets to run 5 times. Table 3 records the average and standard deviation of the test MSE loss. The results prove that our method is statistically significant to outperform the baselines because our confidence interval is always upper than the confidence interval of the baselines. Due to limited computation resources, we do not cover all ten backbones and five datasets, but we believe these results have shown that our method has consistent advantages.

Table 3: The average and standard deviation of MSE in 5 runs

| | BENCHMARKS | | | |
|---|---|---|---|---|
| MODEL | NSE | | SEVIR | |
| | ORI | + SFP | ORI | + SFP |
| CONVLSTM | 0.4092±0.0002 | **0.1277±0.0001** | 0.1762 0.0007 | **0.1279±0.0009** |
| FNO | 0.2227±0.0003 | **0.1007±0.0002** | 0.0787±0.0012 | **0.0437±0.0013** |
| CNO | 0.2192±0.0008 | **0.1492±0.0011** | 0.0057±0.0005 | **0.0053±0.0006** |

## C.3 EXPERIMENTAL SETUP FOR DIRECT PREFERENCE OPTIMIZATION (DPO)

In RQ5, we compare the training stability of SFP against Direct Preference Optimization (DPO). Since DPO is originally designed for discrete language modeling, we adapt it for our continuous spatiotemporal forecasting task as follows:

**1. Preference Data Construction**
For a given input history $x$, we construct a preference pair $(y_w, y_l)$ based on the domain-specific reward function $S(\cdot)$ defined in Appendix B.

- **The Winner ($y_w$):** Defined as either the ground truth future sequence or a model-generated trajectory that yields a high reward score.

- **The Loser ($y_l$):** Defined as a model-generated trajectory that yields a lower reward score, i.e., $S(y_w) > S(y_l)$.

**2. DPO Loss Function**
We treat the pre-trained supervised model as the reference policy $\pi_{\text{ref}}$. The agent (policy) $\pi_\theta$ is optimized to maximize the margin between the likelihood of the winner and the loser, constrained by the KL divergence from the reference policy. The loss function is formulated as:

$$\mathcal{L}_{\text{DPO}}(\pi_\theta; \pi_{\text{ref}}) = -\mathbb{E}_{(x,y_w,y_l)\sim\mathcal{D}} \left[ \log \sigma \left( \beta \log \frac{\pi_\theta(y_w|x)}{\pi_{\text{ref}}(y_w|x)} - \beta \log \frac{\pi_\theta(y_l|x)}{\pi_{\text{ref}}(y_l|x)} \right) \right] \quad (13)$$

where $\sigma$ is the sigmoid function and $\beta$ is a hyperparameter controlling the strength of the KL constraint.

**3. Discussion on Instability**
As discussed in Section 5 (RQ5), we found that optimizing this objective directly in the high-dimensional continuous pixel space often leads to training collapse or mode collapse. In contrast, SFP avoids this issue by decoupling exploration (World Model) from exploitation (Policy Update via pseudo-labels), resulting in superior stability and performance.

