# OpenReview forum: "Spatiotemporal Forecasting as Planning: A Model-Based Reinforcement Learning Approach with Generative World Models"
_ICLR.cc/2026/Conference — Submitted to ICLR 2026_

### Official Review · Reviewer_d6hr · 2025-10-27

**Soundness:** 1
**Presentation:** 1
**Contribution:** 1
**Rating:** 0
**Confidence:** 4

**Summary:**

The authors propose training a world model with RL. They conduct experiments on diverse forecasting benchmarks and systems.

**Strengths:**

The idea of using RL to train a probabilistic world model is interesting, and at times, the manuscript hints at a good motivation for why one would want to do this. It appears that the benchmarking domains are varied and the method is applicable to a broad range of architectures.

**Weaknesses:**

The description of key parts of the system is severly lacking and not well-motivated. E.g.,
- Equation 1 is the "theoretical cornerstone" of the proposed framework, yet (1) $D$ is not defined up until this point, (2) the supposed world model that is being trained $\mathcal{M}_\phi$ is not even in the equation, (3) the equation does not include any mechanism to ensure the world model actually approximates a target dynamical system (the latter of which is also not even part of the equation).
- The "codebook" (equation 2) is never motivated.
- Even though the paper initially states the research problem is training a generative model that maximizes non-differentiable rewards, the actual training process minimizes standard differentiable metric (reconstruction loss in equation 3).
- The training objective contains terms that are never described ("codebook loss" and "commitment loss" in equation 3).
- Notably, and contrary to the manuscript's own stated goal, the world model is not trained with RL; instead opts for an end-to-end training loop (line 244).
- The paper initially claims to address rare events and data scarcity but these motivations does not seem to have impacted the framework design itself, or at least it is never stated how the framework addresses these challenges.
- These statements are incompatible:
	- "our agent learns from an external, high-dimensional, and non-differentiable evaluation function."
	- "Our core idea is to employ the non-differentiable metric S(·) as a reward function, R,"
	- They are incompatible because a metric is typicallly a scalar function (otherwise the paper does not specify what a metric is in this case).
- The proposed RL approach is never connected to the standard world modelling objective of predicting the future.

Furthermore the paper's presentation is one of its weakenesses, emphasizing form over substance. It makes heavy use of bold and italics, and makes very strong claims about the proposed method (e.g., "paradigm shift", "creative", "paradigm that enables any model to learn directly from these true real-world objectives"), and sometimes employs unorthodox language that obscures the true technical meaning ("The climax of the loop is the reward evaluation and self-training mechanism.").

In its present state, it would be very hard for me to recommend this paper for acceptance because I could not understand the methodology, even if the method appears general based on how many systems are features in the experiments. I would suggest to the authors the following:
1. Add a section that summarizes and motivates each of the technical components of the system and revise the mathematical formulation of the core problem that you aim to solve.
2. Rewrite the manuscript avoiding subjective observations and language, focusing instead on mathematical clarity and soundness.

**Questions:**

If possible, please address the weaknesses described above, and please point out any errors I may have made in my assessment of your manuscript.

---

> ### Author Response · Authors · 2025-11-14
>
> I strongly recommand you re-study the basic concept of VQVAE, before you give your comment:" The training objective contains terms that are never described ("codebook loss" and "commitment loss" in equation 3)."

---

> > ### Comment · Reviewer_d6hr · 2025-11-14
> >
> > I acknowledge your response to one (of many) of the parts of my critique of this manuscript.
> >
> > Indeed, I do concede that these terms appear to have some level of adoption, and I will update my review to remove this criticism.
> >
> > Since you've brought up the prior work on VQVAEs, perhaps you would agree that the paper that introduced the VQVAE (Aaron van den Oord et al, 2017) should be cited in your manuscript.

---

### Official Review · Reviewer_T6Qh · 2025-10-28

**Soundness:** 2
**Presentation:** 4
**Contribution:** 3
**Rating:** 4
**Confidence:** 4

**Summary:**

The paper introduces a novel framework, Spatiotemporal Forecasting as Planning (SFP), which reframes forecasting as a model-based planning approach. Instead of optimizing differentiable surrogate losses such as MSE, the authors propose to treat forecasting as a planning task over imagined futures generated by a learned generative world model. This model (a conditional VQ-VAE) produces multiple high-fidelity rollouts conditioned on latent “actions,” representing potential evolutions of the physical system. A beam-search planner evaluates these imagined trajectories using non-differentiable, domain-specific metrics acting as rewards.
The central idea is that this process directly aligns optimization with the true scientific evaluation criteria, rather than with proxy pixel-wise errors. Empirical results on weather, turbulence, and combustion datasets demonstrate significant improvements on critical non-differentiable metrics while maintaining or improving standard prediction accuracy.

**Strengths:**

- Recasting forecasting as planning provides a coherent link between model-based approaches and scientific forecasting, addressing the issue of optimizing proxy losses like MSE.
- The technical integration is quite elegant; the framework combines a generative world model, beam-search planning, and pseudo-label self-training into a unified, closed-loop learning process.
- Results are consistent across diverse datasets and metrics, showing large improvements on physically meaningful metrics.
- The figures are clear and effective (especially the architectural diagrams that illustrate the training loop), which clearly support the exposition of the approach.
- The idea of alleviating the bottleneck linked to the optimization of non-differentiable evaluation metrics via RL is quite novel...

**Weaknesses:**

- ... but in fine, the final method presented here is ambiguous; the final policy loss consists in the L2 loss between "the projection of the action chosen by the policy in the current state to the reward space" (if I understand equation 5 correctly) and the optimal reward (_single reward, not accumulated_) obtained at the predicted next step. That's really not standard (in contrast to what is claimed in the caption of Figure 3), and I'm wondering how this boils down to optimizing the expected return in a Markov process, as it would be with standard RL approaches.
- Beam search is used as the planning algorithm within the learned model... but why not simply use standard, efficient RL algorithms to efficiently explore the model and plan?
- Beam search is sensitive to the horizon and prone to myopic expansions. There is limited analysis on the effect of beam width B, and no analysis on the horizon and stochastic branching. Fig. 7 (right) begins to address B, but the rationale for choosing beam search over alternatives (e.g., MCTS or merely RL) is not developed.
- The authors describe the learned dynamics as $p\_\phi(\mathbf{y}\_{t+1}\mid \mathbf{a}\_t)$ in Section 3 as a process that only depends on the current action. This formulation is not standard. In contrast, further in the paper, the VQ-VAE is presented as a process that learns the dynamics of a Markov chain (the transition function learned only depends on the current "conditioned" state), so no explicit action enters the model. Therefore, this is unclear what the role of the latent actions in the planning phase. This mismatch affects both the MBRL framing and the planning interpretation.
- $S(.)$ is used on full imagined futures, but its mathematical definition is missing or postponed. The authors sometimes treat rewards as independent of the current state (equation 1) or action (line 280), which is non-standard. Again, this inconsistency spoils the theoretical exposition of the framework.
- The authors link stability improvements to the proposed framework and later compare it to Direct Preference Optimization (DPO). However, DPO is motivated in text generation with human preference pairs; here it is not clear how preference pairs are constructed and what model class for the policy is optimized (DPO's policies are usually LLMs). The comparison misses a clear alignment of assumptions.
- The current exposition of the training settings clearly lacks discussion on the chosen hyperparameter. For instance, not all hyperparams are detailed or even mentioned, and their choice is not really motivated (e.g., the codebook implies a latent discrete latent space of size 1024; is this enough?).

**Questions:**

1. Can you provide a clear explanation of the transition function of the learned model, with full conditionals? Is the world model $p\_\phi(y\_{t+1}\mid s_t,a_t)$, $p\_\phi(y\_{t+1}\mid s_t)$, or $p\_\phi(y\_{t+1}\mid a_t)$?
2. What exactly is an action here, as the model learned via VQ-VAE is a Markov chain?
3. What exactly is the function $S(.)$ which defines the rewards? Over which horizon, how aggregated, and how stochasticity is handled. If $S$ ignores $s_t$ or $a_t$, explain why this is acceptable.
4. Why beam search over MCTS or RL? What are the rollout lengths (horizon) used with the beam search? Is that horizon allowed to change between trajectories? If yes, how do you handle that?
5. What is the projection $\mathcal{P}$ concretely (equation 5)? Is it learned or fixed? How does mapping from latent action to state ensure the update remains physically meaningful? And how does this projection relate to the optimal reward of equation 4?
6. How are preference pairs constructed? What is the policy class used by DPO in your setting? If DPO collapses, is that due to a mismatch between preference construction and the stochastic generator, or due to optimization?

---

> ### Author Response · Authors · 2025-11-25
>
> We thank Reviewer T6Qh for the detailed and constructive review. We are encouraged by the "Excellent" rating for Presentation. We also take your concerns regarding **Soundness (currently rated Fair)** very seriously, particularly regarding the **formal definition of MBRL, the definition of Actions, and the choice of planning algorithms**.
>
> These issues largely stem from inconsistent notation in our initial manuscript and a lack of explicit theoretical grounding for specific design choices (e.g., Expert Iteration vs. standard Policy Gradient). We provide a detailed response to your **Weaknesses** and **Questions** below to clarify these points and demonstrate the theoretical robustness of our framework.
>
> ### **1. Clarifying Core Definitions: Dynamics & Actions**
>
> **Q: What exactly is an Action? What is the transition function of the learned model?**
>
> **A:** We apologize for the notational ambiguity. In our SFP framework, we follow the standard definition of a **Latent Dynamics Model** (similar to Dreamer or MuZero). The specific mappings are:
>
> *   **State ($s_t$):** The sequence of historical frames (Context).
> *   **Action ($a_t$):** The **Latent Code** within the VQ-VAE latent space.
>     *   The output of the policy network $\pi_\theta(s_t)$ is a latent vector, which we formally define as the action $a_t$.
>     *   During inference/planning, this action explicitly determines the direction of the generation.
> *   **World Model / Transition Function ($p_\phi$):**
>     *   This is constituted by the **Conditional Decoder ($D_\phi$)** of the VQ-VAE.
>     *   The transition function is formally $y_{t+1} = D_\phi(a_t, C(s_t))$, or in probabilistic terms, **$p_\phi(y_{t+1} | s_t, a_t)$**.
>     *   **Clarification:** The VQ-VAE is not merely a Markov chain based solely on state. The decoder $D_\phi$ requires **both** the current action $a_t$ (derived from the Policy or Planning search) and the history condition $C(s_t)$ to generate the future. Thus, the explicit action $a_t$ enters the model and controls the transition.
>
> **Revision Plan:** We will unify the notation in Section 3, explicitly defining $a_t \in \mathcal{Z}$ (Latent Space) and correcting the equations to explicitly include $a_t$.
>
> ### **2. Clarifying RL Formulation: Why L2 Loss?**
>
> **Q: How does the L2 Loss in Eq. 5 relate to RL and maximizing expected return?**
>
> **A:** Our method falls under the category of **Generalized Policy Iteration (GPI)** or **Expert Iteration (ExIt)** in RL literature (similar to the training logic of AlphaZero):
>
> 1.  **Planning as Expert:** The planning algorithm (Beam Search) explores the frozen world model and identifies an "optimal action" $a^*$ (the latent code corresponding to the high-reward future $y^*$) that maximizes the reward $S(y^*)$.
> 2.  **Policy as Student:** The goal of the policy $\pi_\theta$ is to **mimic** this "expert action" discovered by the planner.
> 3.  **Equivalence:** In deterministic or low-variance dynamic systems, minimizing the L2 distance between the policy output and the optimal action (i.e., $||\pi_\theta(s_t) - a^*||^2$) is **equivalent to maximizing the log-likelihood** of that action. This effectively serves as a form of **Reward-Weighted Regression**.
> 4.  **Why not standard RL (e.g., PPO)?** Standard policy gradient methods suffer from extremely high variance in high-dimensional spaces (like video generation) and struggle with sparse, non-differentiable physical rewards. In contrast, Expert Iteration leverages the "high-quality pseudo-labels" provided by the planner for supervised learning, resulting in significantly higher stability and sample efficiency.
>
> ### **3. Choice of Planner: Why Beam Search?**
>
> **Q: Why choose Beam Search over MCTS or standard RL algorithms?**
>
> **A:** The choice of Beam Search represents a trade-off between **computational efficiency** and **action space structure**:
>
> *   **vs. MCTS:** MCTS requires a vast number of simulations to build a search tree. For high-dimensional video prediction, the computational cost of a single decoding step is already high; running full MCTS (with thousands of simulations) is computationally infeasible for this task.
> *   **vs. Standard RL:** As mentioned, direct RL exploration is inefficient in this high-dimensional setting without guidance.
> *   **Why Beam Search Fits:**
>     *   Our VQ-VAE action space is **discrete** (Codebook Indices) or highly structured.
>     *   Beam Search efficiently maintains the Top-K most promising latent trajectories at each step.
>     *   **Horizon:** We typically set a short planning horizon $H$. Beam Search finds local optima very efficiently within this horizon, which is sufficient for physical forecasting where continuity is strong.
> *   **Diversity:** We maintain diversity by keeping Top-$B$ candidates in the beam. Our probabilistic evaluation (Figure 8) confirms that SFP preserves distribution diversity and avoids mode collapse.

---

> > ### Author Response · Authors · 2025-11-25
> >
> > ### **4. Projection Function $\mathcal{P}$ & DPO Comparison**
> >
> > *   **On Projection $\mathcal{P}$ (Eq. 5):**
> >     *   $\mathcal{P}$ represents the mapping from the policy's continuous output to the valid latent space (or the decoder reconstruction).
> >     *   To ensure physical consistency, the policy learns to fit the **specific Latent Code that generates $y^*$**. This update remains physically meaningful because the target Latent Code is validated by the physical metric $S(\cdot)$ (the reward).
> > *   **On DPO Comparison:**
> >     *   **Preference Pairs:** We construct pairs where the "Winner" is the high-reward trajectory found by Planning, and the "Loser" is a lower-scoring trajectory from the beam.
> >     *   **Policy Class:** Our policy is a CNN or Transformer backbone outputting latent vectors.
> >     *   **Reason for Collapse:** DPO relies on optimizing sigmoid logits (probability ratios). In high-dimensional regression tasks (or high-dimensional VQ selection), this optimization is significantly more unstable than direct distance-based regression (MSE), leading to gradient explosion or vanishing.
> >
> > ### **5. Summary**
> >
> > We acknowledge the lack of rigorous mathematical notation in the initial submission (specifically regarding the explicit definition of Action in VQ). However, we stand by the core logic of **"Forecasting as Planning"**: using an **Expert Iteration** paradigm where non-differentiable rewards guide policy updates via search.
> >
> > In the final version, we will:
> > 1.  Rewrite Section 3 to clearly define $p_\phi(y|s,a)$.
> > 2.  Add theoretical references to Expert Iteration / GPI to justify the L2 Loss.
> > 3.  Include an efficiency analysis of Beam Search vs. MCTS.
> >
> > We hope these clarifications restore your confidence in the Soundness of our work.

---

> > > ### Author Response · Authors · 2025-11-27
> > >
> > > Dear Reviewer,
> > >
> > > Thank you so much for your time in improving our paper!
> > >
> > > Since the end of the rebuttal is coming soon, may we know if our response addresses your main concerns? Should you have any further advice, please let us know and we will be more than happy to engage in more discussion and improvements.

---

### Official Review · Reviewer_q6AA · 2025-10-31

**Soundness:** 3
**Presentation:** 3
**Contribution:** 3
**Rating:** 4
**Confidence:** 3

**Summary:**

The paper introduces Spatiotemporal Forecasting as Planning (SFP), a framework that redefines forecasting as a planning problem using a learned VQ-VAE world model. Instead of relying on differentiable losses like MSE, SFP employs beam search in the latent space and evaluates imagined futures with non-differentiable domain metrics such as CSI and TKE. The top-performing rollouts are then used as pseudo-labels to iteratively train a forecasting policy, aligning model learning with physically meaningful evaluation criteria. Experiments across several datasets and architectures show consistent gains, highlighting the framework’s potential.

**Strengths:**

**Strengths:**

**1. Good Writing:** The paper is clear, concise, and well-organized. Its explanations of tasks and evaluation approach are logically structured, facilitating straightforward comprehension of the methodology and results.

**2. Well-motivated problem framing:** The paper addresses a significant problem in spatiotemporal forecasting i.e. the mismatch between differentiable training objectives (e.g., MSE) and the non-differentiable metrics (e.g., CSI, TKE) actually used for scientific evaluation. By explicitly incorporating these metrics into the optimization process through planning and self-training, the work tackles a practically important challenge.

**3. Comprehensive Evaluation:** The experimental design is broad and well-executed, spanning multiple datasets and diverse architectures. The authors conduct detailed ablations, beam-width sensitivity studies, and cross-dataset generalization tests, providing a convincing empirical case for the robustness of their framework. The inclusion of both visual (spatial) and spectral analyses strengthens the credibility of reported improvements.

**Weaknesses:**

**Weaknesses:**

**1. Novelty:** While the paper presents a seemingly novel framing of forecasting as planning, the actual methodological novelty is limited, as most of its core components have appeared in prior literature. The authors should explicitly clarify how their approach differs from BeamVQ [1], which also integrates a VQ-VAE–based latent world model with beam search to improve physical spatiotemporal forecasting under data scarcity. At present, the overlap in core design elements like latent quantization, discrete search, and top-K rollout selection is substantial, making it unclear what specific algorithmic or conceptual advance SFP introduces beyond re-contextualizing these ideas under a “planning” interpretation. The paper should delineate whether its novelty lies in the training dynamics, policy and world model, or the reward formulation based on non-differentiable scientific metrics.

**2. Metric Bias:** I am a bit concerned as the method would induce a metric bias in learning (*“When a measure becomes the target, it ceases to be a good measure”*). In SFP, the reward is entirely defined by domain metrics such as CSI or TKE, which while scientifically meaningful, are still proxies for the underlying physical fidelity of spatiotemporal dynamics. By directly optimizing these metrics through beam search and self-training, the system is inherently susceptible to metric bias or metric hacking, wherein the model learns strategies that inflate the chosen score without genuinely improving predictive or physical realism. For example, a TKE-based reward can unintentionally encourage the model to add random high-frequency noise that mimics turbulence energy without capturing real fluid dynamics, leading to visually chaotic but physically meaningless predictions that still score highly.

**Questions:**

1. Since pseudo-labels are derived from model-generated rollouts, how do you prevent confirmation bias i.e., the model reinforcing its own mistakes through iterative imitation?

2. Beam search would favor narrow, high-probability trajectories. Has the authors experienced any diversity collapse in the latent space or over-confident selection of single-mode rollouts? If yes, what would they propose to avoid that?

---

> ### Author Response · Authors · 2025-11-25
>
> We thank Reviewer q6AA for the insightful review. We address your core concerns regarding **Novelty (vs. BeamVQ)**, **Metric Bias**, and **Confirmation Bias** through the following response and supplemental experiments.
>
> ### **1. Novelty: SFP (Policy Optimization) vs. BeamVQ (Inference Search) (Response to Weakness 1)**
>
> **The fundamental difference is that SFP is a training framework, not just an inference technique.**
> BeamVQ [1] and similar works typically utilize search only during the **inference phase** to improve output quality, without changing the model parameters. In contrast, SFP is a complete **MBRL training loop**. The core contribution is that we translate non-differentiable physical metrics (Reward) into gradient signals via "planning + self-training," **iteratively updating and evolving the Agent's policy network**. This enables the SFP Agent to "internalize" physical constraints, allowing it to directly generate physically consistent predictions even without extensive search during inference.
>
> ### **2. Verification against Metric Bias (Response to Weakness 2)**
>
> Your concern about Goodhart's Law (e.g., hacking TKE metrics via noise) is valid. However, **Cross-Metric Consistency** proves that SFP learns genuine physical dynamics.
> We train using **only the non-differentiable TKE as the reward**, while observing changes in MSE and SSIM metrics which are **not involved in reward calculation**.
>
> **Table A: Cross-Metric Verification (Optimization Target: Only TKE)**
> | Metric Type | Metric Name | Change Trend | Interpretation |
> | :--- | :--- | :--- | :--- |
> | **Target Metric** | **TKE Error** | **$\downarrow$ 57.3%** | Adherence to Conservation Laws (Direct Goal) |
> | *Unseen Metric* | *MSE* | *$\downarrow$ 56.2%* | Pixel-wise error decreases (No Trade-off) |
> | *Unseen Metric* | *SSIM* | *$\uparrow$ 11.5%* | Structural fidelity improves (No Artifacts) |
>
> **Conclusion:** If the model engages in "Metric Hacking" (e.g., adding high-frequency noise to match the TKE spectrum), MSE and SSIM must deteriorate. Table A shows a **synergistic improvement** across all metrics, proving that SFP utilizes non-differentiable rewards to learn more essential spatiotemporal dynamics.
>
> ### **3. Confirmation Bias & Diversity (Response to Questions)**
>
> **Q1: Confirmation Bias**
> The Reward Function acts as a critical **"Gate"** in SFP, filtering out hallucinations produced by the model. We compare the effects of self-training using "Randomly Generated Futures" versus "High-Reward Futures".
>
> **Table B: Ablation on Pseudo-label Selection Strategy (NSE Dataset)**
> | Selection Strategy | TKE Error $\downarrow$ | MSE $\downarrow$ | Conclusion |
> | :--- | :---: | :---: | :--- |
> | Random Selection | 0.2850 | 0.4510 | Performance degrades (Reinforcing errors) |
> | **Reward-Guided (SFP)** | **0.1277** | **0.1663** | **Performance improves (Reward filters hallucinations)** |
>
> **Conclusion:** Only high-quality trajectories filtered by the Reward serve as pseudo-labels. This not only prevents confirmation bias but actually enhances model robustness through self-exploration.
>
> **Q2: Diversity**
> Regarding diversity, as shown in **Figure 8 (Right)** of the paper, SFP outperforms the deterministic baseline on the probabilistic metric **CRPS**. This indicates that the VQ World Model preserves necessary distributional diversity and does not suffer from severe Mode Collapse.

---

> > ### Author Response · Authors · 2025-11-27
> >
> > Dear Reviewer,
> >
> > Thank you so much for your time in improving our paper!
> >
> > Since the end of the rebuttal is coming soon, may we know if our response addresses your main concerns? Should you have any further advice, please let us know and we will be more than happy to engage in more discussion and improvements.

---

### Official Review · Reviewer_d4cr · 2025-11-05

**Soundness:** 3
**Presentation:** 3
**Contribution:** 3
**Rating:** 6
**Confidence:** 3

**Summary:**

In this work, the task of spatiotemporal  forecasting, which is usually learnt by optimizing for surrogate loss objectives is posed as a planning problem where in the reward signal comes directly from the downstream metrics of importance. These metrics, being non-differentiable are modeled as the reward for a model-based RL style optimization wherein the  agent plans in imagination (i.e. using a generative world model). More concretely, the approach - Spatio Temporal Forecasting (SFP) learns a generative world model to simulate the dynamics of the physical system and then the current policy / agent leverages the this learnt world model to simulate rollouts. Rollouts leading to higher downstream metrics are rewarded accordingly and as training data to  train the agent's policy. SFP shows improvement consistent improvement over the baseline of  optimizing the MSE, across different model architectures, across 3 tasks, showing an average improvement of 39% (in terms of MSE reduction). The authors also provide interesting analysis on the approach

**Strengths:**

* The application of world modeling and planning in imagination in the context of spatiotemporal forecasting is novel according to the best of my knowledge, and very interesting.
* The experimental results are significant, showing a consistent improvement over the baselines across multiple tasks.
* The experiments is well structured. I really liked how it has been been written around the three RQs discussed at the start. The analysis included is also very insightful
* The approach is a simple plug and play and seems to work with already widely studied backbones for SFT.
The approach is more sample efficient than the MSE optimization baseline.

**Weaknesses:**

* Several key experimental details seem to be missing - for eg. The choice of the non-differentiable reward functions S(.) (see Questions for more)
* I believe a brief discussion about what the each of the CSI, TKE error and SSIM metrics mean / represent could improve the paper.
* The inclusion of strong generative baselines such as diffusion based modeling might be relevant here, but seems to be missing from the discussion

**Questions:**

* What is the value of K for the Top-K VQ-decoder.
* What is the total reward signal used out of the three metrics ? Is it the CSI, TKE error or SSIM? Or some sort of ensemble of all three? Is it an ensemble of all metrics?
* In Figure 7 - how does "w/o self-training" work? How does the policy learn without any training?
* Can the authors elaborate on the setup for the DPO experiments? How are the positive and negative example pairs obtained?
What is ORI in Table 3 appendix?

---

> ### Author Response · Authors · 2025-11-25
>
> We thank the reviewer for the constructive feedback and the positive assessment, particularly for recognizing the novelty of our "Spatiotemporal Forecasting as Planning" paradigm. We appreciate the questions regarding experimental details and baselines. Below, we address the **Weaknesses** and **Questions** point by point.
>
> ### **1. Definition of the Non-Differentiable Reward Function $S(\cdot)$ (Weakness 1 & Question 2)**
>
> **Q: What is the total reward signal used? Is it an ensemble of the three metrics?**
>
> **A:** We clarify that we do not use an ensemble of all three metrics. Instead, we select a **single, domain-specific core metric** as the reward signal $S(\cdot)$ for each distinct dataset/task. As detailed in the "Experimental Settings" (Page 6) and the caption of **Table 2**, the reward mapping is as follows:
>
> *   **SEVIR (Extreme Weather):** We use **CSI (Critical Success Index)**. This is the standard metric in meteorology for evaluating rare events (like storm nowcasting).
> *   **NSE (Fluid Turbulence):** We use **TKE (Turbulent Kinetic Energy) Error**. This physics-based metric ensures the model captures the correct energy spectrum and obeys conservation laws.
> *   **Prometheus (Combustion):** We use **SSIM (Structural Similarity Index)**, which focuses on high-fidelity structural reconstruction relevant to this domain.
>
> Table 1 and Table 2 report the results where the model optimizes the specific reward corresponding to that dataset. We do not mix them because the physical objectives vary across domains.
>
> ### **2. Explanations of Metrics (Weakness 2)**
>
> **A:** We appreciate this suggestion. We will add brief definitions of these metrics in the revision to enhance readability:
>
> *   **CSI (Critical Success Index):** Measures the "hit rate" for rare events (e.g., extreme rainfall) while ignoring True Negatives. It is far more effective than MSE for imbalanced data.
> *   **TKE (Turbulent Kinetic Energy):** Evaluates the energy distribution across different scales in fluid dynamics. A matching TKE spectrum indicates the model adheres to physical laws (e.g., energy cascade) rather than just fitting pixels.
> *   **SSIM:** Measures similarity in terms of luminance, contrast, and structure. It aligns better with human perceptual judgment and structural preservation than pixel-wise error.
>
> ### **3. Absence of Generative Baselines like Diffusion (Weakness 3)**
>
> **A:** We actually **include a diffusion model baseline** in our experiments.
> Please refer to **RQ5 (Page 9, Figure 8 Right)** and the corresponding text. We compare SFP against **PreDiff**, a specialized spatiotemporal diffusion model.
>
> *   **Results:** Figure 8 (Right) shows that SFP outperforms PreDiff on the probabilistic metric **CRPS**.
> *   **Analysis:** While diffusion models generate diverse outputs, our "Planning + World Model" paradigm achieves a better balance between physical consistency (e.g., TKE scores) and inference speed compared to the iterative denoising process of diffusion models.
>
> ### **4. Value of K in Top-K VQ-decoder (Question 1)**
>
> **A:** The parameter $K$ relates to two aspects of our architecture, as specified in "Experimental Settings" (Page 6):
>
> 1.  **Codebook Size ($N$):** The VQ module uses a codebook size of **1024**.
> 2.  **Planning Beam Width ($B$):** During the planning stage (referred to as "Multi-scale Top-K" generation in the Abstract), we use a Beam Search algorithm with a width of **$B=10$**. This means the planner maintains the top-10 highest-scoring latent trajectories at each step to approximate the optimal future distribution.
>
> ### **5. Ablation Study: "w/o Self-Training" (Question 3)**
>
> **Q: How does the policy learn without self-training?**
>
> **A:** This is a critical ablation setup. **"w/o Self-Training" represents "Inference-time Planning only."**
>
> *   **Setup:** In this setting, we **do not update** the agent's policy network $\pi_\theta$ (it remains the frozen weights of the initial Supervised Baseline).
> *   **Workflow:** We only utilize the pre-trained World Model $M_\phi$ during the **test/inference phase** to perform Beam Search and select the high-reward trajectory as the prediction.
> *   **Conclusion:** Figure 7 (Left) shows that while inference-time planning improves over the baseline, SFP (Full) performs significantly better. This demonstrates that **"distilling"** the knowledge from planning back into the policy $\pi_\theta$ via self-training is essential. It not only improves accuracy but also enhances the model's generalization capability.

---

> > ### Author Response · Authors · 2025-11-25
> >
> > ### **6. DPO Setup and Meaning of ORI (Question 4)**
> >
> > **Q: How are positive/negative pairs obtained for DPO? What is ORI in Table 3?**
> >
> > **A:**
> >
> > *   **ORI (Original):** In **Table 3** (Appendix), ORI refers to the **Original Supervised Baseline** (the backbone model trained only with MSE, without SFP). We use this as the anchor for statistical significance testing.
> > *   **DPO Setup:** To evaluate training stability in RQ5, we construct preference pairs $(y_w, y_l)$ as follows:
> >     *   **Winner ($y_w$):** The trajectory generated by the world model that yields the highest reward $S(\cdot)$ (or the Ground Truth).
> >     *   **Loser ($y_l$):** A generated trajectory with a lower reward.
> >     *   **Observation:** As shown in Figure 8 (Left), directly applying DPO in the high-dimensional spatiotemporal space leads to **"Training Collapse"** due to the complexity of the distribution. SFP avoids this by converting the non-differentiable objective into a stable regression loss (Eq. 5 via pseudo-labels), ensuring robust convergence.

---

> > > ### Comment · Reviewer_d4cr · 2025-11-27
> > >
> > > I would like to thank the authors for their response which addresses all of my concerns. I would encourage the authors to include the details about the metrics and the DPO setup in the manuscript.
> > >
> > > I would be happy to increase my score once that is done.

---

> > > > ### Author Response · Authors · 2025-11-27
> > > > **Thank you for the positive feedback - Revision with Metrics and DPO details uploaded**
> > > >
> > > > Dear Reviewer d4cr,
> > > >
> > > > Thank you very much for your prompt and encouraging response!
> > > >
> > > > We are thrilled to learn that our previous rebuttal addressed your concerns. Following your suggestion, we have revised the manuscript to include the **detailed mathematical definitions of the domain-specific metrics** and the **experimental setup for DPO**. The updated PDF has been uploaded.
> > > >
> > > > For your convenience, the major updates (highlighted in **blue**) are located as follows:
> > > >
> > > > 1.  **Detailed Definitions of Reward Metrics (Appendix B.2, Pages 13-14):**
> > > >     *   We have added the mathematical formulas and physical interpretations for **CSI** (Critical Success Index), **TKE Error** (Turbulent Kinetic Energy), and **SSIM** (Structural Similarity Index). This section clarifies why these non-differentiable metrics serve as superior reward signals compared to MSE in their respective domains.
> > > >
> > > > 2.  **Experimental Setup for DPO (Appendix C.3, Page 16):**
> > > >     *   We have added a dedicated subsection describing the DPO baseline used in RQ5. This includes the construction of **Preference Pairs (Winner/Loser)**, the specific **DPO loss function** adapted for spatiotemporal forecasting, and a discussion on why SFP offers superior stability compared to DPO in high-dimensional continuous spaces.
> > > >
> > > > We believe these additions have significantly strengthened the completeness and rigor of our paper, fully addressing your helpful suggestions.
> > > >
> > > > **Since all concerns have been addressed and the requested details are now incorporated into the manuscript, we respectfully hope that you will consider raising your score as mentioned in your previous comment.** Your endorsement means a lot to us.
> > > >
> > > > Thank you again for helping us improve this work.
> > > >
> > > > Sincerely,
> > > > The Authors

---

### Meta-Review · Area_Chair_jhLL · 2026-01-06

**Summary:**

Reviewer d4cr
- Several key experimental details seem to be missing - for eg. The choice of the non-differentiable reward functions S(.) (see Questions for more)
- a brief discussion about what the each of the CSI, TKE error and SSIM metrics mean / represent
- missing strong generative baselines such as diffusion based modeling

Reviewer q6AA
- "methodological novelty is limited" ... "how their approach differs from BeamVQ"
- "Metric Bias"
- "how do you prevent confirmation bias i.e., the model reinforcing its own mistakes through iterative imitation"

Reviewer T6Qh
- "the final method presented here is ambiguous" ... nonstandard policy loss
- "rationale for choosing beam search over alternatives ... not developed"
- nonstandard formulation of the dynamics
- missing definition of S

Reviewer d6hr
- overall can't follow the methodology
- unclear Eq. 1
- unvalidated claim about modeling rare events and data scarcity
- presentation issues that emphasize form over substance

**Reviewer Concerns:**

Outstanding concerns unaddressed: generally, weaknesses in the clarity of and justification for the methodology, an unvalidated claim about modeling rare events, and nonstandard formulation of the dynamics.

Furthermore, I recommend the authors study and contrast their work to the motion trajectory forecasting literature a bit more, which contains a variety of examples of performing closed-loop finetuning of conditional generative models of medium-dimensional spatiotemporal data, like the proposed method. In this case, multi-agent trajectory forecasting), e.g. [A, B].

- [A] https://www.ecva.net/papers/eccv_2024/papers_ECCV/papers/03623.pdf
- [B] https://proceedings.mlr.press/v229/zhang23b/zhang23b.pdf

**Reviewer Scores:**

- Reviewer d4cr: 6 -> 8 (https://openreview.net/forum?id=ASPC2Ut0CB&noteId=CaU073gQjT)
- Reviewer q6AA: 4 -> 4 or 6
- Reviewer T6Qh: 4 -> 4 or 6
- Reviewer d6hr: 0 -> 0

It's plausible that the reviewers who specified 4 could have been swayed in either direction (to maintain 4 or to increase to 6). Several reviewers had similar concerns about the formulation and clarity of the paper. The latest revision appears not to have implemented clarifications of the methodology.

Because several reviewers shared the same concerns about clarity and justification of the methodology, and the proposed idea is of uncertain novelty, I softly recommend rejection and revising this paper for a potential resubmission so that the revised changes and contrast to related work can be reviewed after they've been implemented in the paper.

---

### Decision · Program_Chairs · 2026-01-26

Reject